# Synthesis and Antibacterial Activity of Polyalthic Acid Analogs

**DOI:** 10.3390/antibiotics12071202

**Published:** 2023-07-19

**Authors:** Marcela Nunes Argentin, Felipe de Paula Nogueira Cruz, Ariana Borges Souza, Elisa Marcela de Oliveira D’Aurea, Jairo Kenupp Bastos, Sérgio Ricardo Ambrósio, Rodrigo Cassio Sola Veneziani, Ilana Lopes Baratella Cunha Camargo, Cassia Suemi Mizuno

**Affiliations:** 1Laboratory of Molecular Epidemiology and Microbiology, Department of Physics and Interdisciplinary Science, São Carlos Institute of Physics, University of São Paulo, São Carlos 13563-120, SP, Brazil; 2Núcleo de Pesquisa em Ciências Exatas e Tecnológicas, Universidade de Franca, Av. Dr. Armando Salles de Oliveira, 201 Parque Universitário, Franca 14404-600, SP, Brazil; 3School of Pharmaceutical Sciences of Ribeirão Preto, University of São Paulo, Av. do Café S/N, Ribeirão Preto 14040-930, SP, Brazil; 4College of Pharmacy and Health Sciences, Western New England University, Springfield, MA 01109, USA

**Keywords:** polyalthic acid, copaiba oil, diterpene, biofilm, antimicrobial, polyalthic acid analogs

## Abstract

Polyalthic acid (PA) is a diterpene found in copaiba oil. As a continuation of our work with PA, we synthesized PA analogs and investigated their antibacterial effects on preformed biofilms of *Staphylococcus epidermidis* and determined the minimal inhibitory concentration (MIC) of the best analogs against planktonic bacterial cells. There was no difference in activity between the amides **2a** and **2b** and their corresponding amines **3a** and **3b** regarding their ability to eradicate biofilm. PA analogs **2a** and **3a** were able to significantly eradicate the preformed biofilm of *S. epidermidis* and were active against all the Gram-positive bacteria tested (*Enterococcus faecalis*, *Enterococcus faecium*, *S. epidermidis*, *Staphylococcus aureus)*, with different MIC depending on the microorganism. Therefore, PA analogs **2a** and **3a** are of interest for further in vitro and in vivo testing to develop formulations for antibiotic drugs against Gram-positive bacteria.

## 1. Introduction

Antimicrobial resistance (AMR) is an urgent public health problem and has killed nearly 1.27 million [1] people around the world. Every year, there are more than 2.8 million cases of drug-resistant infections [1]. The CDC divides 18 drug-resistant microorganisms into three threat levels: urgent, serious, and concerning [2]. Among the urgent threats are infections by *Clostridioides difficile*, responsible for 223,900 infections and 12,800 deaths per year in the US alone. Infections by this bacterium increased by 400% from 2000 to 2013, and currently, there are only three drugs approved for treating these infections [3]. It was estimated that antimicrobial resistance costs USD 55 billion between medical costs and work absence in the US [4]. Several factors contribute to the high cost of treating drug-resistant infections directly or indirectly. For example, patients with these infections require more extended hospitalizations and more stays in the ICU due to a lack of effective drug treatments. AMR may result in the closing of wards to contain outbreaks, disturbing the hospital’s daily routine [5].

We are currently living in a state of AMR crisis, where the worldwide incidence of drug-resistant infections is increasing without known treatments. This crisis has many causes, including, but not limited to, few antibiotics in the pipeline, antibiotics misuse, inappropriate prescribing, and extensive agricultural use. Pharmaceutical company mergers have decreased the number of companies working on antimicrobial drug discovery. The existing major pharmaceutical companies have discontinued research on antimicrobial discovery due to the high cost of research and increased time to market a drug [6]. Since antibiotics are a short-term treatment, the return on investment is low, compared to long-term treatment drugs such as antihypertensive and antihyperglycemic agents.

Antibiotic development is an important factor in the fight against antibiotic resistance. It has been estimated that by 2050, antimicrobial resistance will be the major cause of death, claiming 10 million lives every year [7]. The need for new antibiotics is urgent and the development of new antibiotics was one of the four core functions proposed during the 2016 UN High-Level Meeting on antimicrobials [8]. However, for reasons already mentioned above, there are not enough new antibiotics in the pipeline. Therefore, basic research conducted at academic institutions, in collaboration with pharmaceutical companies, will be important in contributing to the discovery of new antibiotics [9]. Although about 90% of the antibiotics in clinical use are from actinomycetes [10], plants could also be used to feed the pipeline of new antibiotics due to their rich biodiversity and production of unique and active secondary metabolites.

Copaiba oil is an oleoresin that is a very popular traditional medicine discovered by the natives of the Amazon in Brazil. The oleoresin is most known for its wound-healing properties, which were discovered based on observations of wounded animals’ behavior. Once injured, animals would rub themselves against copaiba trees, causing the release of the oleoresin from the trunk of the tree. Many of its traditional uses have been confirmed in *in vitro* studies, including wound-healing [10], anti-inflammatory [10,11], leishmanicidal [12,13], and antimicrobial activities [14,15].

Copaiba oil’s main constituents are diterpenes (20%) and sesquiterpenes (80%) [11]], and its biological properties are attributed to these compounds. Polyalthic, copalic, hardwickiic, and kaurenoic acids are the most commonly found diterpenes in copaiba oil [16]. While copalic acid is the biomarker for the Copaifera genus, the most studied terpene is kaurenoic acid. Of the several biological effects reported for this compound, its anti-inflammatory activity is noteworthy, being investigated in different animal models. When administered at 3 mg/Kg, kaurenoic acid was able to reduce thermal and mechanical hyperalgesia in mice [17]. Additionally, kaurenoic acid was effective in decreasing swelling in a carrageenan-induced paw edema model [18].

Polyalthic acid (PA, Figure 1) is a far less explored diterpene with reported mild to moderate antibacterial, antifungal, anti-inflammatory, antinociceptive, and cytotoxic activities. Structural modification of this natural product by reacting the carboxylic acid with amines of different sizes and shapes has improved its antileishmanial activity [19]. Therefore, in the search for new effective antibiotics against resistant strains, we describe the synthesis of PA analogs and their activity against biofilm formation and growth inhibition of Gram-positive and Gram-negative bacteria.

## 2. Results 

Four PA analogs (**2a**, **2b**, **3a**, and **3b**) were synthesized, as depicted in Figure 1, in the search for new molecules with antibacterial activity. The reaction of PA with triphenylphosphine, carbon tetrachloride, and amines produced the amides **2a** and **2b** (Appendix A). Reduction of the amides with DIBAL-H generated amines **3a** and **3b** (Appendix A). Compounds **2a** and **2b** have been previously reported, but 3a and 3b are novel. The investigation of the potential antimicrobial activity of the PA and its analogs started with the biofilm assay. All five compounds were able to reduce the formed biofilm (Figure 1) significantly. PA was able to reduce the preformed biofilm by 64%. Compounds **2a** and **3a** were the most active, decreasing the biofilm by 97% (Figure 1). Based on the biofilm eradication activity, the most active compounds, **2a** and **3a**, were tested for growth inhibition of Gram-positive and Gram-negative bacteria, and their activity was compared to PA using the broth microdilution assay. PA was only active against *S. epidermidis* at 256 mg/L. By kinetic analysis, we observed differences in the slope of the curve with and without PA at the higher concentration tested in the remaining species analyzed (Appendix A). Therefore, the PA MIC for the other species were shown as >512 mg/L. The susceptibility of the bacterial strains used in this study against currently approved antibiotics are shown in Appendix A. PA and its analogs were tested against several multidrug-resistant strains. Compounds 2a and 3a were active against all five Gram-positive bacteria (Figure 2) but not active against Gram-negative bacteria (Figure 3) when tested at 512 mg/L. These results are in line with the results reported by other groups [20], including Urzúa et al. [21], in which 15 diterpenes were tested for their antimicrobial activity, and all of them were active against Gram-positive bacteria only. 

The MIC of the compounds **2a** (Appendix A) and **3a** (Appendix A) against different Gram-positive bacteria was determined by observing the kinetic growth revealed by its metabolism using the Biolog Redox Dye mix H 100× (Biolog, Cat#74228). Both compounds were active at concentrations ≤32 mg/L (Table 1; Appendix A). Compound **3a** showed the best activity (MIC = 8 mg/L) against *E. faecalis*. The activity of amide **2a** and amine **3a** varied depending on the microorganism. For *S. aureus* ATCC 25923 and *E. faecium*, the MIC was the same (16 mg/L). This activity against *E. faecium* is lower than that reported for PA (IC_50_ = 8.5 µg/mL) [22]; however, the strain used in this study was different. Amine **3a** was more active against *S. epidermidis* and *E. faecalis* than the corresponding amide. For *S. aureus* ATCC 8095, amide **2a** was more active than amine **3a** (16 and 32 mg/L, respectively). The minimal bactericidal concentration (MBC) could not be calculated since the tested compounds formed a suspension with the broth. Hence, the turbidity of the broth could interfere with the reading of the results. Since the MBC/MIC ratio could not be calculated, it was not possible to determine if the compounds were bactericidal or bacteriostatic.

To determine the safety of the compounds, the hemolytic activity of PA, **2a**, and **3a** was evaluated (Figure 4). No significant hemolytic activity was observed for all three compounds, even at the highest concentration of 512 mg/L.

The hemolytic activity was similar for PA and its analogs. Amide **2a** was a little less hemolytic than the corresponding amine **3a** (7.5% versus 8.2%), but all three compounds were much less hemolytic than the positive control triton X (C+).

## 3. Discussion

Bacteria have developed resistance to nearly all antibiotics in clinical use. One of the methods by which bacteria become resistant to antibiotics is through biofilm formation. Biofilms work as a protective barrier, impermeable to antibiotics, allowing bacteria to survive and grow. PA and its four analogs were active in the biofilm eradication assay. Based on the results, the activity depended more on the type of substituent (piperazine (**2a**, **3a**) versus 2-aminoethoxy) ethanol (**2b**, **3b**)), since there was not much difference in activity between the amides and amines (**2a** × **3a** and **2b** × **3b**). PA had its antibiofilm activity tested against several oral pathogens that cause dental caries and endodontic infections. The MICB_50_ ranged between 3.12 and 50 µg/mL, with the best activity shown *against Lactobacillus casei*, with an MICB_50_ of 3.12 µg/mL [23]. Here, we demonstrated that PA was able to reduce the preformed biofilm produced by *S. epidermidis,* and it was more active than **2b** and **3b**. Analogs **2a** and **3a** almost completely eradicated the biofilm when tested at the same concentration. This is the first study reporting the antibiofilm activity of PA and its analogs against *S. epidermidis*. These results are significant because this bacterium is a multidrug-resistant strain (Appendix A). The diterpene andrographolide, which is similar in structure to PA, decreased *L. monocytogenes* biofilm formation by 73% when tested at 1 mg/mL [24]. Comparison between the activities of these compounds is not possible since the studies used different bacteria and we tested the compounds at only one concentration (512 mg/L).

Since the PA analogs **2a** and **3a** significantly decreased the preformed biofilm of *S. epidermidis* (by approximately 97%), we decided to investigate whether they would also show antimicrobial activity against *S. epidermidis* and other relevant bacterial species.

In 2017, the WHO published a priority list (critical, high, and medium) of drug-resistant pathogenic bacteria in need of new antibiotics [25]. Based on this list, four Gram-negative bacteria (*K. pneumoniae*, *E. coli*, *A. baumannii*, and *P. aeruginosa*) and five Gram-positive bacteria (*S. epidermidis*, *S. aureus*, *E. faecalis*, and *E. faecium*) were selected to test PA and its analogs **2a** and **3a**. Although the compounds were not active against any of the Gram-negative bacteria tested, **2a** and **3a** were active against all Gram-positive bacteria tested, including some multidrug-resistant Gram-positive strains such as *S. epidermidis* and *E. faecium* (Table 1 and Appendix A). Vancomycin-resistant *E. faecium* is on the Priority 2 (high) list of bacteria for which new antibiotics are urgently needed. The compound with the best activity was **3a**, which was active against *E. faecalis*. The MICs of the compounds were higher when compared to the positive control daptomycin (Table 1). However, the terpene pharmacophore is attractive because it has shown promising effects in the treatment of resistant strains. It has been reported that some terpenes can modulate antibiotic resistance [26,27], and terpenes do not usually cause resistance [28]. In previous studies, PA was bactericidal against several Gram-positive bacteria, with the lowest MIC of 6.25 mg/L against *P. gengivallis and P. micros* [23]. Except for *S. epidermidis*, PA was unable to inhibit the bacteria metabolism in our study, even at the highest concentration tested (Table 1). However, we noticed a difference in the slope of the curve (Appendix A), which might indicate that at higher concentrations, PA could be active against the other Gram-positive bacteria. PA analogs demonstrated much better activity than PA, showing that structural modification of the natural product significantly improved antimicrobial activity. 

The lack of activity against Gram-negative bacteria may be associated with the compound’s difficulty in penetrating the outer membrane of these microorganisms. The tested compounds were not soluble and formed a suspension in cation-adjusted Mueller Hinton Broth 2 (CAMHB), even with the addition of 2% DMSO. Thus, broth filtration was not possible since it could remove the tested PA analogs. Therefore, a control (negative control) consisting of the broth plus the PA analogs, without the bacteria, was made to detect contamination of the broth. A reduction of the dye is indicative of contamination. Daptomycin (DAP) was used as an antibiotic control, and its MIC was determined against the Gram-positive bacteria used in this study (Table 1 and Appendix A). As shown in Figure 2 and Figure 3, bacterial growth can be observed in the growth control group (broth plus bacteria) but could not be detected in the negative control (broth plus tested compound), suggesting that there was no bacterial contamination of the broth.

The antimicrobial mechanism of action of PA is not known. However, studies of the mechanism of action of diterpenes suggest that their antimicrobial activity is associated with their ability to alter the physical properties of the bacterial membrane, affecting its function. Magic-angle spinning nuclear magnetic resonance (MAS-NMR) studies of the diterpenes (+) totarol [29] and abietic acid [30] revealed that these diterpenes are located on the upper region of the membrane. This location reinforces earlier findings about the mechanism of action of (+) totarol, which is thought to incorporate itself into the membrane and disturb phospholipids’ packing [31] and membrane function. It has been proposed that the interaction between diterpenes and the membrane may alter the respiratory chain, causing cell death [32]. Considering this mechanism of action, lipophilicity is an important factor for activity. Indeed, a linear relationship between diterpenes’ antimicrobial activity and lipophilicity has been reported [20,21]. Another factor affecting the antimicrobial activity of diterpenes is the presence of hydrogen-bond-donor (HBD) groups. Generally, active diterpenes have one HBD, and compounds having two HBDs are less active. Due to the limited number of compounds, it was not possible to establish a structure activity relationship (SAR) for the PA analogs.

Some natural products, such as triterpenoids and steroid saponins, have hemolytic activity [33]. The amphiphilic character of saponins allows them to interact with cell membranes and disrupt the membrane [34]. Kaurenoic acid, another diterpene found in copaiba oil, has shown significant hemolytic activity against human and animal erythrocytes in a dose-dependent manner [34,35]. 

Despite its promising activity against resistant bacterial strains, one of the obstacles preventing terpenes from entering clinical trials is its toxicity. These molecules can disrupt cell membrane integrity, leading to cell lysis and death [35]. However, the results of the hemolytic assay showed that PA, **2a**, and **3a** have low toxicity to human blood cells, suggesting that it could be tolerated by the body, even at high doses.

In conclusion, this study described the synthesis of four PA analogs (using PA as the starting material) and their antimicrobial and hemolytic activities. Analogs **2a** and **3a** were able to reduce almost completely the biofilm formed by *S. epidermidis*, with a better activity than PA. Both compounds **2a** and **3a** were also active against Gram-positive planktonic bacteria and were better than PA. These results are encouraging because some of the compounds were active against multidrug-resistant bacterial strains. The compound with the best activity was **3a**, with an MIC of 8 mg/L against *E. faecalis* ATCC 29212. Although other tests are needed, the safety of the compounds was evaluated in this study through hemolytic assay and showed that PA, **2a**, and **3a** display very low hemolytic activity. Thus, PA could provide a unique template for new antibacterial drugs active against resistant bacterial strains. Further investigations are required to elucidate the mechanism of action of the analogs.

## 4. Material and Methods

Polyalthic acid was isolated as described previously [19].

Polyalthic acid (PA). ent-15,16-epoxi-8(17),13(16),14-labdatrien-18-oic; ^1^HNMR (CDCl_3_, 400 MHz): d 0.72 (s, 3H, C_20_-Me); 1.15 (s, 3H, C_19_-Me); 4.60 (s, 1H, C_17_-H); 4.89 (s, 1H, C_17’_-H); 6.26 (dd, 1H, J = 1.6 and 0.8 Hz, C_14_-H), 7.20 (s, 1H, C_16_-H); 7.35 (t, 1H, J = 1.6 Hz, C_15_-H); ^13^C NMR (CDCl_3_, 100 MHz): d 15.1, 16.3, 18.4, 23.5, 24.0, 26.8, 37.1, 37.8, 37.9, 38.7, 47.4, 49.5, 56.1, 107.1, 111.0, 125.4, 138.7, 142.8, 147.8, 185.6.

### 4.1. General Procedure for the Amide Synthesis

The synthesis of amides was performed according to published procedures [36]. In a 50 mL round-bottom flask, a mixture of PPh_3_ (10 equiv) and CCl_4_ (10 mL) was stirred under reflux temperature for 5 h in an inert atmosphere. The solution was allowed to cool to room temperature and polyalthic acid (1 equiv) dissolved in CCl_4_ was added. The solution was heated under reflux for 30 min. The solvent was removed under reduced pressure and the amine (20 equiv) dissolved in dichloromethane (5 mL) was added. The mixture was stirred overnight at room temperature. Subsequently, the solvent from the mixture was removed under vacuum, and the crude mixture was dissolved in ethyl acetate. The resulting solution was washed with 1N HCl and saturated solution of NaHCO_3_. The organic phase was dried over MgSO_4_ and the solvent was removed. The crude product was purified through automated flash chromatography eluted with hexanes/ethyl acetate.

((1S,4aS,5R)-5-(2-(furan-3-yl)ethyl)-1,4a-dimethyl-6-methylenedecahydronaphthalen-1-yl)(4-methylpiperazin-1-yl) methanone **2a**.

Reaction of polyalthic acid (200 mg, 0.63 mM) with 1-methylpiperazine (1402 µL, 12.6 mM) afforded 160 mg (63% yield) of **2a**; ^1^HNMR (CDCl_3_, 300 MHz): δ 0.70 (s, 3H, C_20_-Me); 1.16 (s, 3H, C_19_-Me); 2.22 (s, 3H, N-Me); 2.31 (t, 4H, J= 4.5 Hz, CH_2_-N amine); 3.52–3.68 (m, 4H, CH_2_-N amide); 4.53 (s, 1H, C_17_-H); 4.81 (s, 1H, C_17’_-H); 6.19 (s, 1H, C_14_-H); 7.13 (s, 1H, C_16_-H); 7.27 (s, 1H, C_15_-H); ^13^C NMR (CDCl_3_, 75 MHz): δ 15.0, 18.7, 19.2, 23.6, 24.0, 26.9, 35.8, 37.6, 38.1, 39.3, 45.9, 46.0, 46.9, 49.9, 55.2 (3C), 56.4, 106.8, 110.9, 125.4, 138.7, 142.6, 148.0, 177.3. LC/MS m/z % 399 (100) [M + H]^+^.

(1S,4aS,5R)-5-(2-(furan-3-yl)ethyl)-N-(2-(2-hydroxyethoxy)ethyl)-1,4a-dimethyl-6-methylenedecahydronaphthalene-1-carboxamide **2b.**

Reaction of polyalthic acid (200 mg, 0.63 mM) with 2- (2-aminoethoxy) ethanol (1268 µL, 12.6 mM) afforded 139 mg (54% yield) of **2b**; ^1^HNMR (CDCl_3_, 300 MHz): δ 0.69 (s, 3H, C_20_-Me); 1.11 (s, 3H, C_19_-Me); 3.38–3.46 (m, 2 H, CH_2_-OH); 3.51–3.55 (m, 4H, CH_2_-O-CH_2_); 3.70 (m, 2H, CH_2_-N); 4.54 (s, 1H, C_17_-H); 4.82 (s, 1H, C_17’_-H); 6.22 (s, 1H, C_14_-H), 7.16 (s, 1H, C_16_-H); 7.31 (s, 1H, C_15_-H); ^13^C NMR (CDCl_3_, 75 MHz): δ 15.0, 16.6, 18.6, 23.5, 23.9, 26.2, 37.2, 37.9 (2C), 39.0, 39.5, 47.4, 50.1, 56.0, 61.7, 70.0, 72.3, 106.8, 111.0, 125.5, 138.7, 142.7, 147.9, 179.0. LC/MS m/z % 404 (100%) [M + H]^+^.

### 4.2. General Procedure for the Amine Synthesis

Amide (1 equiv) was dissolved in anhydrous THF (10 mL) and the solution was cooled to 0 °C. DIBAL-H (8.5 equiv) was added. The solution was stirred under an inert atmosphere for 4 h. A mixture of methanol and water (10:0.5) was added. THF was removed under vacuum and the aqueous phase extracted three times with dichloromethane. The organic phase was combined and dried over MgSO_4_, and the solvent was removed. The crude product was purified through automated flash chromatography eluted with hexanes/ethyl acetate or chloroform/methanol.

1-(((1R,5S,8aS)-1-(2-(furan-3-yl)ethyl)-decahydro-5,8a-dimethyl-2-methylenenaphthalen-5-yl)methyl)-4-methylpiperazine **3a**.

Reaction of amide **2a** (181 mg, 0.47 mM) with DIBAL-H (4.5 mL, 25% in toluene, 5.64 mM) in THF afforded 64 mg (33% yield) of amine **3a**. δ 0.69 (s, 6H, C_19–20_-Me); 2.28 (s, 3H, N-Me); 4.54 (s, 1H, C_17_-H); 4.83 (s, 1H, C_17’_-H); 6.24 (s, 1H, C_14_-H); 7.18 (s, 1H, C_16_-H); 7.32 (s, 1H, C_15_-H); ^13^C NMR (CDCl_3_, 75 MHz): δ 15.1; 18.8 (2C); 23.7; 24.2; 24.3; 36.6; 38.0; 38.7; 39.1; 39.4; 45.8 (2C); 49.6; 55.4 (2C); 55.6; 56.4; 68.8; 106.4; 110.9; 125.6; 138.5; 142.5; 148.3. LC/MS m/z % 385 (40%) [M + H]^+^.

2-(2-(((1R,5S,8aS)-1-(2-(furan-3-yl)ethyl)-decahydro-5,8a-dimethyl-2-methylenenaphthalen-5-yl)methylamino)ethoxy)ethanol **3b**.

Reaction of amide **2b** (370 mg, 0.916 mM) with DIBAL-H (6.49 mL, 25% in toluene, 7.79 mM) in THF afforded 48 mg (13% yield) of amine **3b**; ^1^HNMR (CDCl_3_, 300 MHz): δ 0.70 (s, 3H, C_20_-Me); 0.79 (s, 3H, C_19_-Me); 3.56 3.62 (m, 4H, CH_2_-O-CH_2_); 3.62–3.71 (m, 2H, CH_2_-N); 4.54 (s, 1H, C_17_-H); 4.84 (s, 1H, C_17′_-H); 6.24 (s, 1H, C_14_-H); 7.17 (s, 1H, C_16_-H); 7.32 (s, 1H, C_15_-H); ^13^C NMR (CDCl_3_, 75 MHz): δ 14.9; 18.7; 19.3; 23.4; 24.0 (2C); 36.3; 37.0; 37.9; 36.4; 39.4; 50.0; 50.2; 55.9; 61.6; 61.8; 69.8; 72.1; 106.3; 110.9; 125.5; 138.5; 142.5; 148.1. LC/MS m/z % 390 (100%) [M + H]^+^.

### 4.3. Biofilm Eradication Assay

A single colony of *S. epidermidis* ATCC 12228 (negative control; poor biofilm former) and *S. epidermidis* ATCC 35984 (positive control; good biofilm former) was taken from the overnight culture on brain heart infusion agar (BHIA), transferred to a 0.75% glucose-supplemented brain heart infusion (BHI) broth, and incubated at 37 °C for 18 h under agitation. Cultures were adjusted to an OD_600_ of 0.9 to 1.1 in phosphate-buffered saline (PBS), transferred to flat bottom 96-well polystyrene microplates, and incubated at 37 °C for 24 h to allow biofilm formation according to Qin et al. 2014 [37] with few modifications. Planktonic cells were removed by washing with PBS. The biofilm eradication capacity of the compounds was tested in the biofilm formed by *S. epidermidis* ATCC 35984 in 12 replicates. The compounds were diluted at 512 mg/L in fresh BHI broth, adding 1% DMSO when necessary, and incubated under the same conditions. This second step allowed the compound to disperse the formed biofilm. Biofilm staining was made by adding 0.2% crystal violet with 15 min incubation step at room temperature, followed by three washes with PBS. Finally, the crystal violet-stained biofilm was solubilized with ethanol:acetone (80:20) solution, which was transferred to another microplate for indirect quantification of the biofilm at 595 nm using Spectramax M5 (Molecular Devices, San Jose, CA USA). The mean absorbance of the *S. epidermidis* ATCC 35984 biofilm was compared to that of *S. epidermidis* ATCC 12228 biofilm, both without any compound, using a Student’s *t*-test, and a *p*-value < 0.05 indicated a significant difference between the means of the two different strains, guaranteeing the quality of the biofilm formed by *S. epidermidis* ATCC 35984. To evaluate the compound’s biofilm eradication capacity, data analysis was performed based on the comparison of the final biofilm produced by *S. epidermidis* ATCC 35984 without any compound and then with each compound. The replicates’ mean was used and the standard deviation was calculated to analyze the coherence between the replicates. Analysis of variance (ANOVA) of the mean absorbance of biofilm produced by the *S. epidermidis* ATCC 35984 without any compound and then with each compound was performed; *p*-value < 0.05 indicated a significant difference between the two values.

### 4.4. Screening Test

The screening assay was performed according to published procedures [38]. A 100-times concentrated stock solution of each test compound was prepared by diluting the compounds in DMSO. Then, the stock solution was diluted 1:100 in cation-adjusted Mueller-Hinton Broth (CAMHB) (BD, East Rutherford, NJ, USA) containing 1.05% Biolog Redox Dye Mix H (100×) (Biolog, Hayward, CA, USA) (CAMHB-H). Each compound was tested at a concentration of 512 mg/L using 96-well microplates. As positive controls, each bacterium without the test compound was added to the CAMHB to observe the growth. As negative controls, CAMHB-H was added to a well to exclude media contamination, and CAMHB-H with the test compound at 512 mg/L without bacteria was added to another well to exclude contamination due to the compound presence. Any bacterial growth in the CAMHB-H causes a reduction in the Biolog Redox Dye Mix H and the media color changes from yellow to purple. All microplates were incubated at 36 °C ± 1 °C for 24 h using OmniLog (Biolog, USA), which has a CCD camera to capture the color variation in OmniLog units (OUs) every 15 min using Biolog’s OmniLog Data Collection v. 3.0 software. This data was used to plot the growth kinetic of each well using Origin^®^ 2020 v.9.7.0.185 (OriginLab) software. After incubation, the plates were also visually examined for bacterial growth. A positive screening result indicated the antibacterial activity at 512 mg/L with a possible MIC of 512 mg/L or less (≤512 mg/L). A negative screening meant that bacterial growth was observed, indicating no antibacterial activity at the tested concentration. However, a negative screening result at this concentration does not rule out the possibility of antibacterial activity at higher compound concentrations. Therefore, the result was reported as “>512 mg/L”. All tests were performed in triplicate.

### 4.5. Determination of the Minimal Inhibitory Concentration (MIC)

Determination of the susceptibility profile of the strains tested in this study against antibiotics currently in use in hospitals was performed using Phoenix M50 (BD, East Rutherford, NJ, USA) according to the manufacturer’s recommendations and using the panels NMIC/ID-470 BD and PMIC/ID-89 for Gram-negative and Gram-positive bacteria, respectively. Results were interpreted following EUCAST recommendations.

The MIC was determined using the broth microdilution method [39,40] against all bacterial species strains that demonstrated susceptibility in the screening test. A 100-times concentrated stock solution of each compound was prepared by diluting the compounds in DMSO. Then, the stock solution was diluted 1:100 in cation-adjusted Mueller-Hinton Broth (CAMHB) (BD, East Rutherford, NJ, USA) containing 1.05% Biolog Redox Dye Mix H (100×) (CAMHB-H). Each test compound was diluted to a final concentration of 512 mg/L, from which a serial dilution (1:2) was performed so it could be tested in concentrations ranging from 512 mg/L to 0.06 mg/L. As positive controls, each bacterium without the test compound was added to the CAMHB to observe the growth. As negative controls, CAMHB-H was added to a well to exclude media contamination, and CAMHB-H with the test compound in different concentrations without bacteria was added to wells to exclude contamination due to the compound presence. Any bacterial growth in the CAMHB-H causes a reduction in the Biolog Redox Dye Mix H, and the media color changes from yellow to purple. All microplates were incubated at 36 °C ± 1 °C for 24 h using OmniLog (Biolog, USA), which has a CCD camera to capture the color variation in OmniLog units (OUs) every 15 min using Biolog’s OmniLog Data Collection v. 3.0 software. This data was used to plot the growth kinetic of each well using Origin^®^ 2020 v.9.7.0.185 (OriginLab) software. By observing the kinetic graphics, the MIC is the lowest concentration of the test compound to inhibit bacterial growth. All tests were performed in triplicate.

### 4.6. In Vitro Assessment of Hemolytic Properties

The ability of compounds to lyse red cells was evaluated based on the method described by Santos-Filho et al. [41] in which venous blood samples obtained from healthy donors were collected and stored in vials containing ethylenediaminetetraacetic acid (EDTA). Subsequently, the erythrocytes were precipitated and resuspended in 1% PBS after three serial washes in PBS and incubated in 96-well round bottom microplates containing the target compounds at 512 mg/L for 1 h at 37 °C. Triton X-100 1% was used as positive control.

The plates were then centrifuged at 800× *g*/15 min at room temperature and the supernatant was analyzed at an OD of 405 nm. The hemolysis percentage and the experimental error were calculated from the ratio of the absorbance of target compound and positive control 100×, compound standard deviation, and hemolytic percentage, respectively. The assays were performed in technical duplicates and analyzed using GraphPad Prism 5.0 software.

## Data Availability

The data presented in this study are available in the Appendix A.

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
