# Peer review of "Synthesis and Antibacterial Activity of Polyalthic Acid Analogs"

_antibiotics, 2023, doi:10.3390/antibiotics12071202_

Round 1

Reviewer 1 Report

Argentin et al. describe the preparation of four random derivatives of polyalthic acid and their evaluation for their antimicrobial properties. The paper has serious flaws. Firstly, the depicted structures are incorrect and do not show acid or amide, but instead peroxide and hydroxylamine, respectively. The number of tested molecules is very insufficient (definitely for drawing any conclusions regarding their SAR, e.g. effect of lipophilicity). There is no comparison of the activity of derivatives to the activity of the lead compound (PA) or in many cases to any positive control. Therefore, in reviewer’s opinion, the work does not meet the scientific criteria and should be rejected.

Author Response

Dear reviewer,

Please see the authors reply to your comments in the attached document.

Reviewer 2 Report

The main question addressed by the research is synthesis and Antibacterial Activity of Polyalthic Acid Analogs agisnt the films of Staphylococcus epidermidis and determined the minimal inhibitory concentration (MIC) of the best analogs against planktonic bacterial cells.

The topic is original. However, the rationale, major gap evident in the literature and how the present study contribute a new knowledge are not clear. In fact, what is the major gap in the literature is not clearly mentioned.

The novelty is not clear when compared with the published results.

The scheme for the synthesis is not clear and not properly illustrated. The test and control experiments are not discussed properly and no conclusion were drawn based on these aspects. The conclusion is not consistent. It should be inline with the results. The uniqueness and novelty should be clear. The potential demerits of the study should be incorporated.

The discussion section should be strengthened with the latest studies in the literature.

Table 2 should include more details. It looks very simple without contributing much. The control experiments are not clear there.

The language must be improved

Author Response

Dear reviewer,

Please see the author's reply to your comments in the attached document.

Thanks,

Cassia Mizuno

Reviewer 3 Report

In an antimicrobial resistance context worldwide, authors aimed to evaluate antibacterial activity of synthetized polyalthic acid (PA) analogs in order to propose new antimicrobial drugs. For this purpose, authors first, described the synthesis of the 4 PA analogs. Then, they evaluated their activity against biofilm formation and growth inhibition of gram-positive and gram-negative bacteria.

Overall, it is a well-designed study, and the topic is indeed interesting and scientifically exciting area of research particularly in the current context of antimicrobial resistance worldwide. The state of the art and research objectives were clearly presented in the introduction. The experiment was correctly designed and described in the materials and methods section and the data treatment was appropriate.

In my opinion, the manuscript can be suitable for publication after further changes. I would like to point out some comments that the authors should consider to improve their manuscript.

How authors determined the dose used 512 mg/mL for biofilm eradication test? Authors should explain this point.

I also have a major concern as far as concerns these PA analogs:

To determine whether these PA analogs could be used as antimicrobial drugs, their activity should be compared to reference antibiotics. Have the authors done this? The authors should at least discuss this point. In my opinion, it is a crucial point to assess the “value” of these analogs.

Author Response

Dear reviewer,

Please see the author's reply to your comments in the attached document.

Thank you,

Cassia Mizuno

Reviewer 4 Report

The manuscript is worthy of publication.

Author Response

Dear reviewer,

Please see the author's reply to your comments in the attached document.

Thank you.

Cassia Mizuno

Round 2

Reviewer 1 Report

Reviewer appretiates, that the postive control was added to the investigation. However, nothing changes on the revievewrs opinion, that reporting four random analogues, without any justification of the logic of the structures and with a moderate activity is not a suitable material for publishing in a scientific journal with an impact factor of 5. Moreover, eventhough the attempts to fix the structures of the molecules were carried out, this is not, how properly drawn molecule looks like (the amides). The overall scientific soundness is also very low, and therefore, reviewer does not recommend this article for publishing.

Author Response

Please see the author's response in the attached document.

Reviewer 2 Report

The authors have addressed the comments.

need minor revision in English editing

Author Response

The authors have proofread the manuscript.

Reviewer 3 Report

I went through the revised version of the manuscript. Authors have adressed all the Issues raised and manuscript has been highly improved. The manuscript can be accepted in its present form.

Author Response

The authors thank the reviewer for the time and suggestions to make this paper better.